# Solution-processed perovskite light emitting diodes with efficiency exceeding 15% through additive-controlled nanostructure tailoring

Muyang Ban[1], Yatao Zou[1], Jasmine P.H. Rivett[2], Yingguo Yang[3], Tudor H. Thomas[2], Yeshu Tan[1], Tao Song[1], Xingyu Gao[3], Dan Credgington [2], Felix Deschler[2], Henning Sirringhaus[2] & Baoquan Sun [1]

Organometal halide perovskites (OHP) are promising materials for low-cost, high-efficiency light-emitting diodes. In films with a distribution of two-dimensional OHP nanosheets and small three-dimensional nanocrystals, an energy funnel can be realized that concentrates the excitations in highly efficient radiative recombination centers. However, this energy funnel is likely to contain inefficient pathways as the size distribution of nanocrystals, the phase separation between the OHP and the organic phase. Here, we demonstrate that the OHP crystallite distribution and phase separation can be precisely controlled by adding a molecule that suppresses crystallization of the organic phase. We use these improved material properties to achieve OHP light-emitting diodes with an external quantum efficiency of 15.5%. Our results demonstrate that through the addition of judiciously selected molecular additives, sufficient carrier confinement with first-order recombination characteristics, and efficient suppression of non-radiative recombination can be achieved while retaining efficient charge transport characteristics.

[1] Jiangsu Key Laboratory for Carbon-Based Functional Materials & Devices, Institute of Functional Nano & Soft Materials (FUNSOM), Joint International Research Laboratory of Carbon-Based Functional Materials and Devices, Soochow University, 199 Ren'ai Road, Suzhou 215123, People's Republic of China. [2] Cavendish Laboratory, Department of Physics, University of Cambridge, JJ Thomson Avenue, Cambridge CB3 0HE, UK. [3] Shanghai Synchrotron Radiation Facility (SSRF), Shanghai Institute of Applied Physics, Chinese Academy of Sciences, 239 Zhangheng Road, Pudong New Area, Shanghai 201204, China. These authors contributed equally: Muyang Ban, Yatao Zou, Jasmine P. H. Rivett. Correspondence and requests for materials should be addressed to F.D. (email: fd297@cam.ac.uk) or to H.S. (email: hs220@cam.ac.uk) or to B.S. (email: bqsun@suda.edu.cn)

Organometal halide perovskites (OHPs) are emerging as promising materials for light-emitting diodes (LEDs) due to their high photoluminescence (PL) efficiency, high color purity (full-width at half-maximum, FWHM approximately 20 nm), facile solution processability, and tunable bandgap[1–10]. In a LED under electrical bias, equilibrium is established between bound excitons and free charge carriers. A larger exciton binding energy ($E_b$) promises faster radiative recombination, and thus more efficient electroluminescence (EL). Unfortunately, three-dimensional (3D) OHP displays a small $E_b$ (tens meV), leading to slow electron–hole capture rates[10,11]. Therefore, spatial confinement of electron and hole densities is required in a 3D OHP film to promote radiative recombination[3]. Lower dimensional (such as two-dimensional (2D) or nano-crystalline) OHP counterparts, with dimensions approaching the Bohr diameter, can enhance $E_b$ through quantum confinement[12]. A 2D Ruddlesden–Popper OHP ($L_2A_{n-1}Pb_nX_{3n+1}$) comprises different layers of [$PbX_6$] octahedra sandwiched between ammonium halide (L) barrier layers. The quantum confinement gradually enhances as the thickness of the $L_2A_{n-1}Pb_nX_{3n+1}$ layer reducing. In order to achieve this, different large ammonium halides (L, for example, PEA = phenylethylammonium; BA = n-butylammonium; NMA = 1-naphthylmethylamonium) are generally used to impede $APbX_3$ crystal growth (A = Cs, $CH_3NH_3^+$ or HC$(NH_2)_2^+$, X = Cl, Br, I)[4–8].

In such low-dimensional structures, besides quantum confinement, there is also dielectric confinement arising from the large difference in the dielectric constants of the organic ligands ($\varepsilon_{org}$) and the inorganic phase ($\varepsilon_{inorg}$)[12]. This leads to exciton wave functions being tightly confined in the 2D inorganic layers with a large $E_b$ (up to hundreds of meV)[13]. In such OHP films an ensemble of self-organized nano-crystalline domains with different excitonic energy reflecting the distribution of sizes around an average $n$ is typically created in low-dimensional perovskite films[4,5]. Such an inhomogeneous excitonic energy distribution can funnel energy towards the nano-crystalline domains with the lowest bandgap. Recently, a high external quantum efficiency (EQE) up to 11.7% has been achieved via incorporating NMAI to achieve such an energy funnel effect[5]. However, both the size and spatial distribution of OHP nanocrystals are random and it is likely that the energy funnel follows inefficient pathways in the current poorly controlled microstructures. What is urgently needed now to further improve device performance are methods for better controlling the distribution of nanocrystals and enhancing radiative yields of low-dimensional OHP materials.

Here, we present an approach for controlling the nanoscale domain structure of thin OHP films. We investigate cesium lead bromide ($CsPbBr_3$) films in which we control domain size and film morphology through the addition of PEABr. Surprisingly, we discover that when incorporating even small concentration of an organic molecule 1,4,7,10,13,16-hexaoxacyclooctadecane (referred to as 'crown' in the following sections, the molecule structure is shown in Supplementary Figure 15) as an additive we are able to suppress the crystallization of the PEABr phase and achieve an improved domain size distribution and more controlled phase separation between the organic and inorganic phase. This results in a significant improvement in the photoluminescence quantum yield (PLQY) up to approximately 70% and allows us to achieve an EQE of 15.5%, which to the best of our knowledge is the highest performance reported for an OHP LED to date. We report the photo- and device physics that is responsible for this significant improvement in device performance.

## Results

**Characterization of film microstructure.** We fabricate our films by spin-coating precursor solutions comprising lead bromide ($PbBr_2$), cesium bromide (CsBr), and PEABr in dimethyl sulfoxide (DMSO) (see Methods section). The concentration of PEABr, i.e. the molar ratio of $M_{PEABr}/M_{PbBr_2} = x\%$, plays an important role in determining the film morphology. As shown by the scanning electron microscope (SEM) and atomic force microscope (AFM) images in Fig. 1a, where perovskite films without PEABr (0%) exhibit a discontinuous, 3D crystallite morphology. OHP films with 20% PEABr display much improved film quality associated with reduced perovskite crystallite size, as shown in Fig. 1b. This suggests that perovskite crystal growth is impeded in the presence of PEABr. The crystallite size is reduced further with increasing the PEABr ratio up to 40%, as shown in Fig. 1c. The root mean square roughness (RMS) extracted from AFM images decreases from approximately 20 nm (0% PEABr) to approximately 1.7 nm (40% PEABr). This result is consistent with previous reports of $MAPbBr_3$ films, for which BABr addition can dramatically enhance the film uniformity[6]. The impeding effect of PEABr on perovskite crystal growth can be ascribed to the strong hydrogen bond between the hydrogen atom of PEABr and the halide atom in $PbBr_6$(ref. [14]) (Supplementary Note 1 and Supplementary Figure 1).

In addition, a preferential orientation of crystallites with respect to the substrate is induced by adding PEABr[4,15]. Both synchrotron 2D grazing incidence XRD (GIXRD) (Fig. 1 and Supplementary Figure 4) and $\theta$–2$\theta$ XRD (Supplementary Figure 3) reveal that the crystal (k00) diffractions become preferentially oriented normal to the substrate with increasing PEABr concentration, while the pristine (0% PEABr) films exhibit powder-like rings in the diffraction pattern indicating a random crystallite orientation (Supplementary Note 2). This 2D OHP preferential orientation may be ascribed to the formation of self-stacked 2D nanoplatelets[4,15]. These nanoplatelets can be observed in SEM images for the perovskite (5% and 10% PEABr) films, as shown in Supplementary Figure 2. In addition, analysis of the width of the XRD diffractions also confirms that the OHP crystallite size is strongly reduced with incorporating PEABr. The FWHM of the XRD peaks become wider with increasing amount of PEABr. The average nanostructure size extracted from (100) peaks by the Scherrer equation decreases from approximately 42.8 nm (0% PEABr) to approximately 14.3 nm (60% PEABr) (Supplementary Table 1), which is consistent with the AFM and SEM measurements. Both GIXRD and $\theta$–2$\theta$ XRD patterns provide evidence for crystallization of the PEABr phase (Supplementary Note 2). PEABr diffraction peaks become particularly prominent if the PEABr ratio is over 40% in Fig. 1c, d (arrow pointed), Supplementary Figures 3 and 4 (labeled as peak '*') when its ratio is over 40% (Supplementary Note 2).

We anticipate that in order to better control phase separation and the distribution of crystallite sizes it would be helpful to suppress the PEABr crystallization. To achieve this, we explore the addition of a small amount of crown molecules. 2D GIXRD and XRD data (Fig. 1e and Supplementary Figures 3 and 4) confirm that crown does indeed inhibit PEABr crystallization. The characteristic PEABr diffraction peaks disappear upon the addition of crown. The SEM and AFM images (Fig. 1e) show smaller crystallite size and smoother film morphology with addition of crown. The crystallite average size decreases from approximately 18.5 nm (40% PEABr) to approximately 13.3 nm (40% PEABr-crown) as determined from the width of the XRD (100) peaks. Both SEM and AFM images (Fig. 1e) corroborate this finding, showing reduced crystallite size in the plane of the

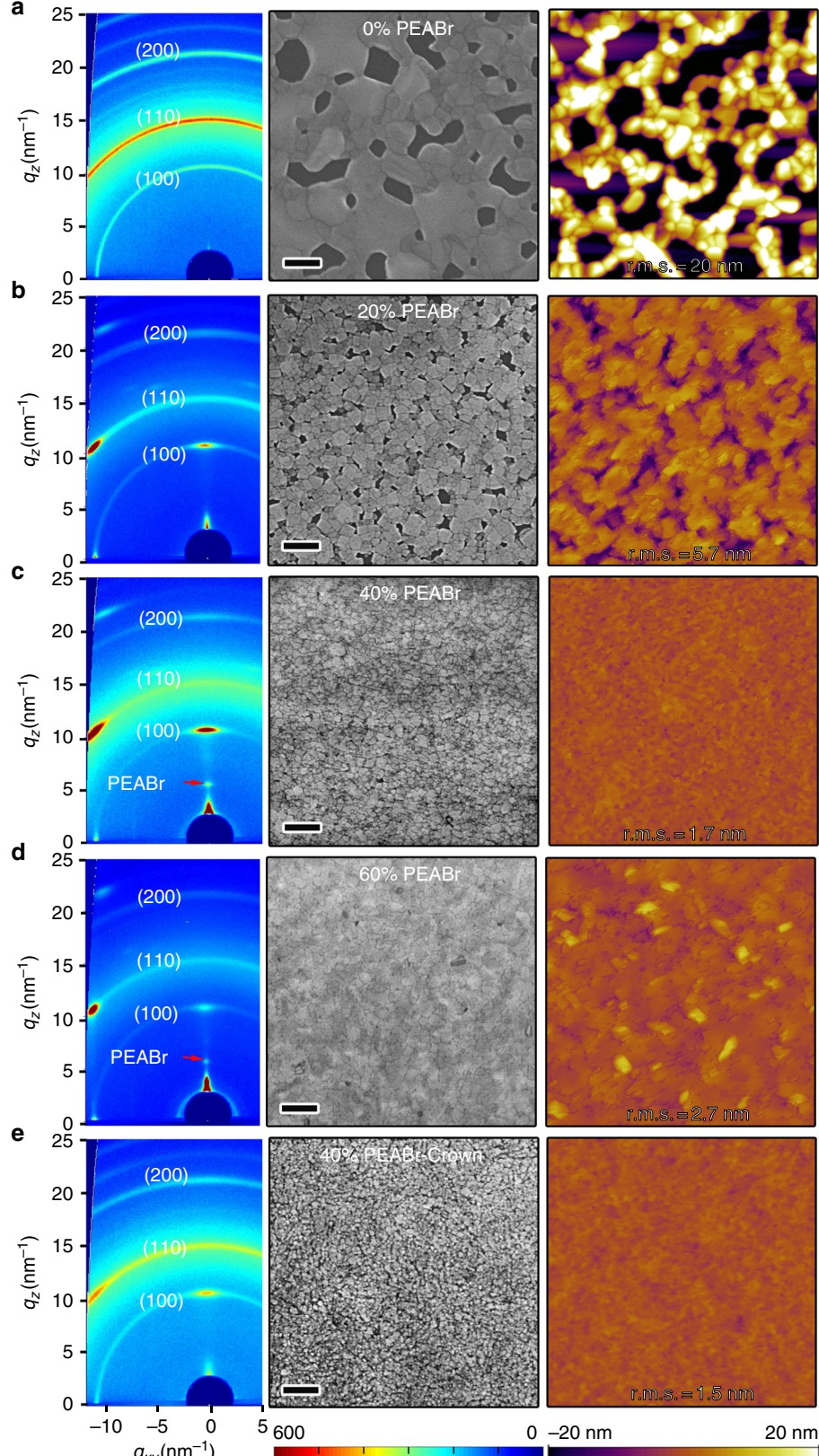

**Fig. 1** 2D GIXRD and morphology study of the perovskite films. 2D GIXRD patterns (left), SEM (middle), and AFM (right) images of the perovskite films with: **a** 0% PEABr, **b** 20% PEABr, **c** 40% PEABr, **d** 60% PEABr, and **e** 40% PEABr-crown. The PEABr molar ratio of x% PEABr means $M_{PEABr}/M_{PbBr_2} = x\%$. The red arrows in the 2D GIXRD images highlight new diffraction peaks of PEABr at the high PEABr concentration. The scale bar of SEM images is 200 nm. The scan area of all AFM images is 2 μm × 2 μm

sample. Furthermore, both XRD and GIXRD diffraction data show evidences for reduced preferential orientation upon adding crown. The 2D GIXRD pattern of the crown sample has a more pronounced ring-like and less textured appearance than the samples without crown, as shown in Fig. 1e. The (110) peaks is observed again in XRD (Supplementary Figure 3) and the azimuthally integrated scattering intensity of different GIXRD profiles along the ring at $q = 10.7\ nm^{-1}$ (100) plane displays reduced preferential orientation (Supplementary Figure 5) for 40% PEABr-crown sample (Supplementary Note 2). Cartoon images show these perovskite orientation tendencies with PEABr or PEABr-crown, as shown in Supplementary Figure 6.

The π–π stacking between PEA cations can be suppressed due to steric hindrance when incorporating crown. Both TEM and SEM-energy-dispersive X-ray spectroscopy images confirm this suppressed PEABr aggregation and phase segregation (Supplementary Figures 7 and 8). In the presence of crown, the PEABr aggregation is dramatically suppressed. The ammonium head of PEA cation can form multiple hydrogen bonding with the six oxygen atoms of crown (N–H⋯O)[16]. This interaction is also probed by [1]H nuclear magnetic resonance (NMR) spectra (Supplementary Figure 14 and Supplementary Note 4). Proton resonance signals of crown (peak at $\delta = 3.506$ p.p.m.) shift downfield after incorporation of CsBr (3.529 p.p.m.), PbBr$_2$ (3.531 p.p.m.), and PEABr (3.547 p.p.m.), respectively. This observation is consistent with previous report on downfield chemical shift of proton in similar crown ether for [1]H NMR spectra because of the interaction between crown and cation[17]. And this downfield chemical shift for proton in crown increases from Cs$^+$, to Pb$^{2+}$ and PEA$^+$, which indicates that the interaction between crown and PEA$^+$ is the strongest[17]. We also see evidences for such molecular interactions already in solution: in dynamic light scattering (DLS) experiments (Supplementary Figure 15 and Supplementary Note 4), solutions of CsBr, PbBr$_2$, and PEABr comprise large, scattering particles with size of several hundred nanometers, which we attribute to the formation of PEABr aggregates. Besides suppressing the formation of these aggregates, crown can also suppress the growth of perovskite crystals due to its interaction with Pb$^{2+}$ and Cs$^+$ ions, as shown in Supplementary Figures 9–13 and Supplementary Note 3. As the crystallite nucleation in the precursor solution directly affects the microstructure and morphology of the OHP film, our observation provides further evidences that the addition of crown leads to a more controlled nucleation of crystallites on the substrate, resulting in smaller OHP crystallites and less phase segregations in the OHP films.

**Optical properties of photo-excitations**. We investigate the effects of material structure on excited states from optical absorption and PL experiments (Fig. 2a, b). With increasing concentration of PEABr, the optical absorption edge around 520 nm as well as the peak of the PL emission spectrum display blueshifts which is ascribed to the decrease of average perovskite crystals size. The $x = 0$% films exhibit an exciton-like absorption peak at approximately 520 nm, which is commonly observed in the 3D bulk film[17]. This peak gradually disappears with the increasing PEABr ratio. With increasing PEABr concentration we observe the appearance of three pronounced, new absorption peaks at 405, 436, and 467 nm, which we attribute to the formation of a substantial fraction of 2D PEA$_2$Cs$_{n-1}$Pb$_n$Br$_{3n+1}$ nanoplatelets in the film, and a reduction of the bulk phase[18]. This is consistent with the observed concomitant gradual loss of the bulk exciton peak at approximately 520 nm. The absorption peaks can be ascribed to $n = 1, 2,$ and 3 nanoplatelets[19]. The

concentration of these nanoplatelets in the films increases with PEABr concentration growing. It is interesting to note that we do not see PL emission from these nanoplatelets. In all films the emitted PL locate at wavelengths longer than 500 nm, which indicates that the emission comes almost completely from larger 3D perovskite nanocrystals or thick nanoplatelets ($n > 15$)[20] and that very efficient energy transfer occurs from the small nano-platelets to these larger crystallites, as shown in Fig. 3c. This point is also confirmed by following transient optical measurements. The PL peaks gradually shift from 528 nm (0% PEABr) to 507 nm (100% PEABr), which indicates a shift of the distribution of average crystallite size to smaller values with the increasing PEABr ratio. These observations indicate that an ensemble of nano-crystalline OHP with different sizes and excitonic states are formed upon PEABr addition[4,5].

It is important to note that the average size of our PEABr OHP nanocrystals (over 10 nm) is still significantly larger than Bohr diameter of CsPbBr$_3$ (approximately 7 nm)[21], suggesting that quantum size confinement is unlikely to be the main mechanism for the increased PLQY. There is, however, also dielectric confinement to consider arising from the different dielectric constants of the two components, i.e., the small $\mathcal{E}_{org}$ of PEABr and the large $\mathcal{E}_{inorg}$ of the perovskite[22]. To realize effective dielectric confinement, it is required to form a uniform morphology in which the inorganic perovskite crystallites are well dispersed in the organic matrix of PEABr[22]. If crystallite of the perovskite aggregate and phase separates from the organic phase, dielectric confinement effects are inhibited. For $x \geq 60$% PEABr films, AFM measurements indicate increased RMS roughness of approximately 2.7 nm in comparison with 40% one, suggesting possible PEABr aggregation, which is also consistent with the XRD measurements (Supplementary Figure 3 and Supplementary Note 2). These observations suggest that PEABr impedes perovskite crystal growth and leads to smaller crystallites and improved film morphology. At higher PEABr concentrations the strong π–π conjugation between the planar rings of PEABr encourages formation of PEABr crystals that enhance the driving force for an undesirable, larger-scale phase separation between the perovskite, and the organic phase, thus weakening dielectric confinement effects and introducing non-radiative recombination channels.

PEABr incorporation significantly increases the PL efficiencies (Fig. 2c). While for the 0% PEABr film a very low PLQY (blow 1%) is measured, the PLQY of the 40% PEABr film reaches a much higher value of 23 ± 5%. This suggests that the presence of 2D nanoplatelets, the ensuing confinement effect, and the formation of excitons with large $E_b$ (>100 meV) suppresses the non-radiative recombination processes that limit the PLQY of the 0% PEABr film. Further increase of the PEABr concentration beyond 40% leads to a decrease in PLQY, which may reflect the distribution of crystallite sizes and inefficiencies in the energy transfer funnel[4]. With the addition of crown (dashed lines in Fig. 2a) the absorption features due to PEA$_2$Cs$_{n-1}$Pb$_n$Br$_{3n+1}$ nanoplatelets below 500 nm are reduced, particularly when the PEABr ratio is below 60%. The addition of crown above 20% PEABr leads to a further, large enhancement of the PLQY (Fig. 2c). The 40% PEABr-crown films exhibit a PLQY of 70 ± 8%, which is close to previous record yields for hybrid perovskites[5]. The PL lifetime (1/$e$) increases from 3 to 8 ns upon addition of crown (Fig. 2d), which suggests a reduction of non-radiative decay channels. The PL peak of the 0% PEABr film shows blueshift with adding crown, reflecting a reduction in crystal size, while for the films comprising PEABr, addition of crown leads to a small redshift of the PL peak. This cannot be explained as size effect as the structural characterization discussed above, which

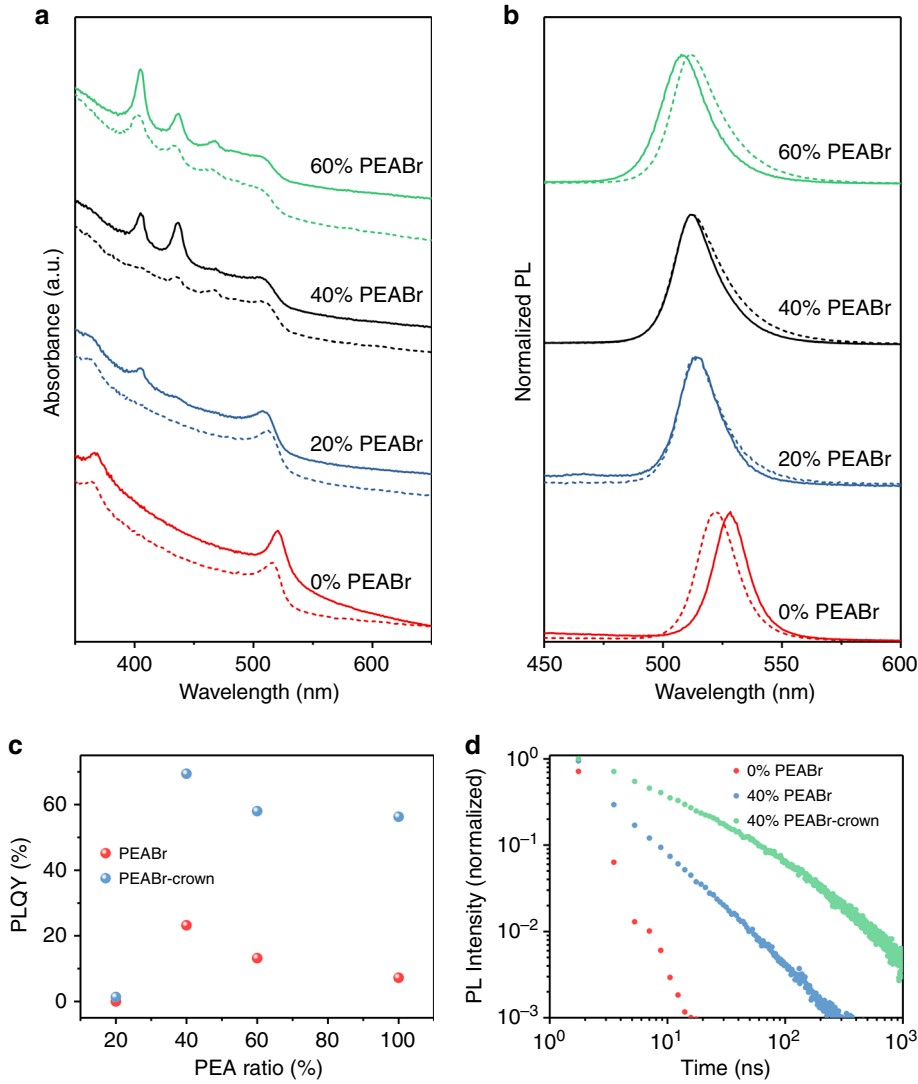

**Fig. 2** Absorption, PL, PLQY, and time-resolved PL of the perovskite film. **a** Optical absorption and **b** PL for the perovskite films without (solid lines) or with (dashed lines) crown for different PEABr ratios. **c** PLQY (excited at 365 nm at approximately 2.3 mW cm$^{-2}$) versus PEABr ratio for the perovskite with and without crown. **d** Time dependence of PL intensity for the perovskite films with 0% PEABr (red), 40% PEABr (blue), and 40% PEABr-crown (black)

clearly indicates a reduction (not an increase) in crystallite size when adding crown. However, this might be a manifestation of the more small-scale phase separation leading to more pronounced dielectric confinement effects as discussed above, an average increased $E_b$ is expected, as shown in Supplementary Figure 17 and Supplementary Note 6. The increased $E_b$ is consistent with enhanced mono-molecule recombination for 40% PEABr with crown, which will be discussed later. A large $E_b$ should result in reduced optical bandgap ($E_{opt.}$); therefore, a redshift in PL peaks is observed. Temperature-dependent PL shows that the 40% PEABr-crown film displays the largest $E_b$. In addition, the strongest low-energy state PL intensity in 40% PEABr-crown film at low temperature hints that its 2D platelet dimensional distribution is significantly different from 40% PEABr one (Supplementary Figure 16 and Supplementary Note 5). We note that the observed improvement of PLQY with addition of crown is not confined to the present system, but can also be observed in other OHP films (Supplementary Figure 18, Supplementary Table 2 and Supplementary Note 7).

**Perovskite LEDs**. A cross-sectional SEM image of an LED device with a bottom hole-injecting layer of poly [bis (4-phenyl) (4-butylphenyl) amine] (poly-TPD) on indium tin oxide (ITO) and a top electron-injecting layer of 2′,2′-(1,3,5-benzinetriyl)-tris(1-phenyl-1-H-benzimidazole)) (TPBi)/LiF/Al and a corresponding energy diagram are shown in Fig. 3a, b, respectively. The current density–voltage–luminance ($J$–$V$–$L$) curves and normalized EL spectra of LEDs with different concentrations of PEABr are shown in Supplementary Figure 19 (Supplementary Note 8). The EL intensity for 0% PEABr CsPbBr$_3$ films is too weak to be measured, reflecting the poor film morphology and low luminescence efficiency. With the increasing PEABr molar ratio, the leakage current in the range of 1–3 V decreases to $10^{-6}$ mA cm$^{-2}$ and the luminance increases up to a maximum of 7000 cd m$^{-2}$ at $x = 40\%$. At even higher concentration ($x = 60\%$) leakage currents increase again and luminance decreases reflecting the poorer film morphology and reduced PLQY. The EL peak is blueshifted from 514 to 510 nm, which is in line with the PL measurement. However, all devices with only PEABr exhibit relatively poor LED performance; a maximum EQE of 1.25% is

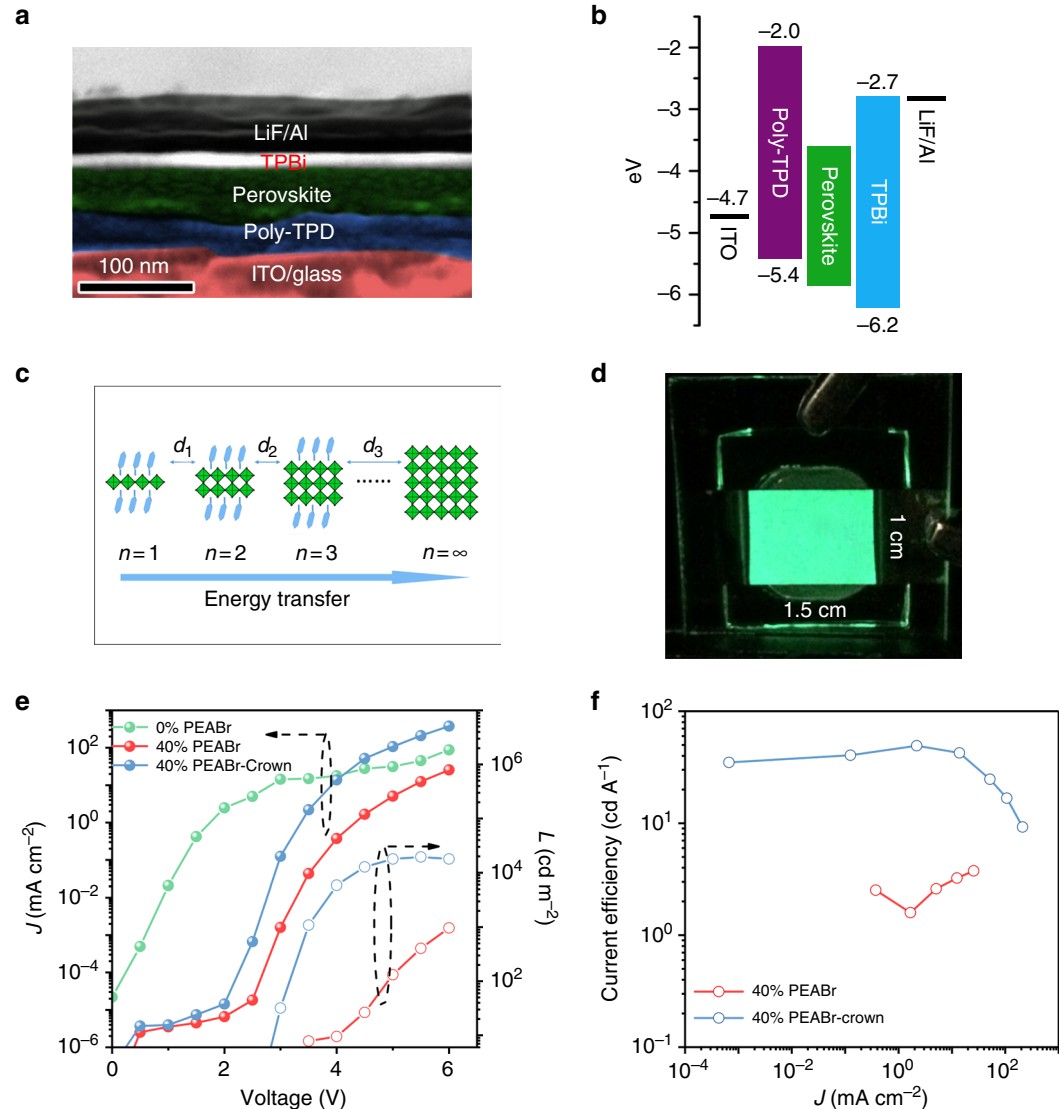

**Fig. 3** Performance characterization of the LED devices. **a** SEM cross-sectional image. **b** Energy diagram of the LED device structure. In the energy diagram, the bandgap energy difference between 2D and 3D is not considered. The energy levels of organic semiconductor layers are taken from literature[6]. **c** A cartoon image shows the energy funnel transfer from thin layer with large bandgap (small $n$) to thick layer with small bandgap (larger $n$), the distances ($d$: $d_1$, $d_2$, ..., $d_n$)) between them play a key role on funnel transfer efficiency. **d** A photograph of operating 40% PEABr-crown LED device with an emitting size of $1.5 \times 1$ cm$^2$ driven at a bias of 3 V. **e** $J$–$V$–$L$ data and **f** current efficiency of devices based on the perovskite films 0% PEABr, 40% PEABr, and with 40% PEABr-crown

achieved for the film with 40% PEABr. However, with incorporating of crown, the LED performance is dramatically improved. While the leakage current remains low, the EL turn-on voltage is lowered, the current density is increased by an order of magnitude and the luminance by two to three orders of magnitude upon incorporation of crown, as shown in Fig. 3e. Figure 3d displays that the EL emission from the LED is uniform and the emission can be assumed as a standard Lambertian profile (Supplementary Figure 20a). A large-area device with a size of $1.5 \times 1$ cm$^2$ was made to test the uniformity of emission. It exhibits bright and large-area uniform emission, as shown in Supplementary Figure 20b. The EQE (measured by the setup shown in Supplementary Figure 21) reaches a maximum value of 15.5% and the current efficiency (CE) reaches 49.1 cd A$^{-1}$, as shown in Fig. 3f. To the best of our knowledge, this is the highest performance for an OHP LED reported to date, EQE is improved by approximately 30% over the highest performance reported

previously (Supplementary Table 3). Device performance statistics for 42 devices are shown in Supplementary Figure 22, and the average EQE is 12.5%.

We observe some hysteresis effects in the current or luminance versus voltage curves of the LEDs, which are assigned to residual ion migration and associated trap states[1]. This is shown for the 40% PEABr perovskite LED with and without crown in Supplementary Figure 23. The level of hysteresis is comparable with previously reported results and is sufficiently minor that it does not affect the LED performance characterization[6]. It is interesting to note that the 40% PEABr-crown device exhibits reduced hysteresis. We believe that this is consistent with the more efficient carrier confinement and suppression of non-radiative recombination which may be manifestations of a reduced trap density. Supplementary Figure 24 shows the scanning-rate-dependent CE for 40% PEABr-crown device. It indicates that the CE is dependent on scanning rate[6]. A tentative

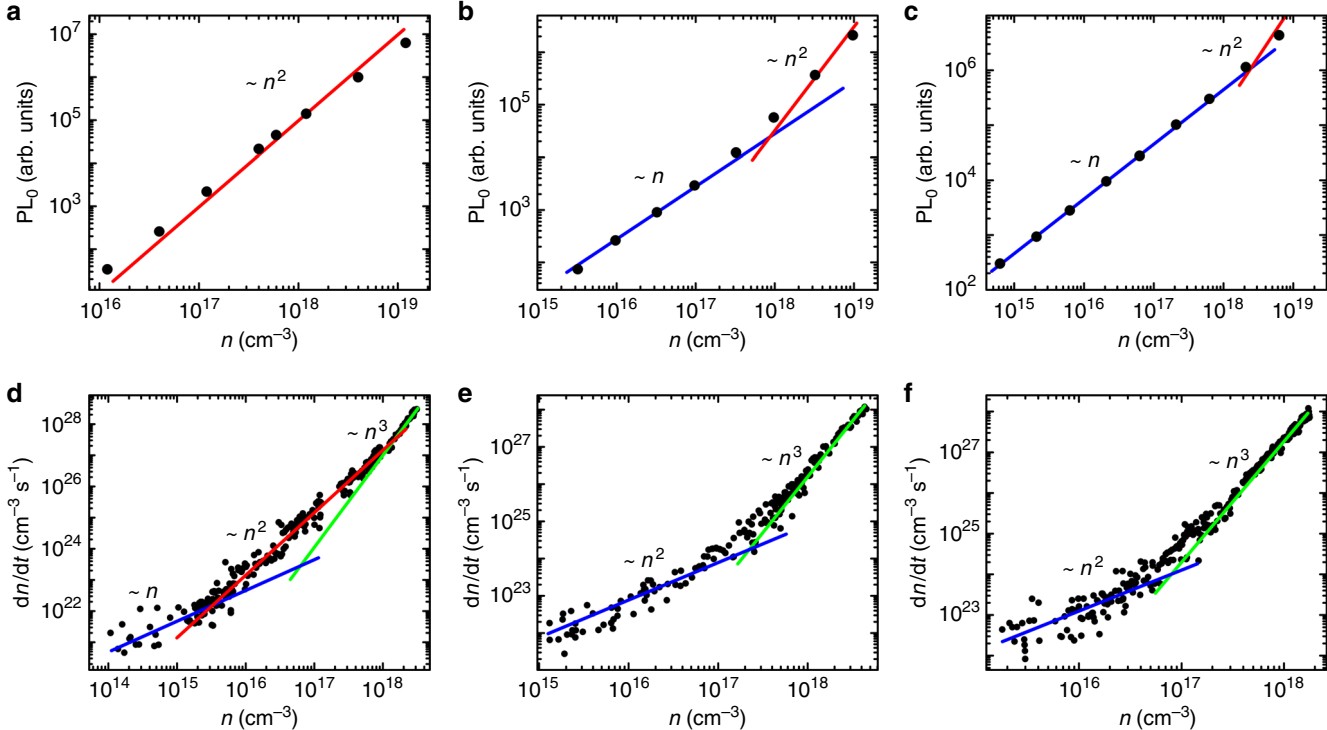

**Fig. 4** Radiative recombination in the perovskite films. PL intensity immediately after photoexcitation (PL$_0$) (400 nm wavelength, 50 fs pulse width) plotted against optical excitation density ($n$) for perovskite films with **a** 0% PEABr, **b** 40% PEABr, or **c** 40% PEABr-crown. Recombination rate ($dn/dt$) plotted against $n$ for perovskite films with **d** 0% PEABr, **e** 40% PEABr, or **f** 40% PEABr-crown. Blue, red, and green lines are guides for the eye representing linear, quadratic, and cubic dependencies respectively

mechanism for the observed device degradation is Joule heating when the step size of the measurement is too small.

In addition to the high performance, the PEABr-crown device also shows improved stability. With a constant voltage of 3.5 V operation, the PEABr-based device decrease to approximately 70% of its initial value while the PEABr-crown one remains almost constant in 100 s (Supplementary Figure 25a). In addition, the EQE of the PEABr-crown based device drops to half of its initial value after approximately 90 min of constant current density operation (Supplementary Figure 25b), which is comparable to the reported perovskite LEDs (Supplementary Table 4). In contrast, PEABr-based devices degrade more quickly and it only takes approximately 60 min for their EQE to drop to half of its initial value. We speculate that the inferior device operation stability may be correlated with the poor thermal stability of OHP perovskite when Joule heat is generated during device operation. Perovskites with higher thermal stability would be expected to show improved operational lifetime.

It is interesting to compare this device performance to that of state-of-the-art, synthetic CsPbBr$_3$ nanocrystals that have smaller diameters than the nano-crystalline domains present in our films and can achieve PLQY of 100%[23]. However, the best LEDs fabricated to date with such nanocrystals in the same device architecture have EQE values of only approximately 5.0% (Supplementary Figure 26 and Supplementary Note 9). This reflects the poor charge transport properties of such synthetic crystallite LEDs, in which the long ligands required to stabilize the nanocrystals in solution inhibit the charge transport properties so severely that ultrathin films (below 20 nm) are needed to achieve optimum device performance. In our precursor-based films PEABr provides a shorter and less insulating ligand, and efficient charge transport can be realized even in relatively thick (approximately 50 nm) films.

**Transient photoexcitation dynamics**. To study the luminescent and recombination processes in the OHP films, transient PL and transient absorption (TA) measurements were conducted (Fig. 4). PL$_0$ is the intensity of emission immediately after excitation (time-zero). As such, it directly measures the radiative recombination rate at a given excitation density. For 0% PEABr CsPbBr$_3$, PL$_0$ increases quadratically with excitation density, which we assign to bimolecular recombination of non-geminate free electrons and free holes[11]. For the PEABr-based OHP both with and without crown, PL$_0$ increases linearly with excitation density, indicating that PL originates from recombination of photoexcited geminate electron and hole pairs. This is consistent with the formation of weakly-coupled nanocrystals which confine the geminate electron-hole pair. Only at very high excitation densities, PL$_0$ scales quadratically with density, which indicates the emergence of higher-order recombination processes. We attribute this to the formation of multiple excitations in the nanocrystallite, and potentially free carriers, due to screening at high excitation densities, which leads to the onset of second-order radiative and third-order Auger recombination. Normalized PL decays at different excitation densities are plotted in Supplementary Figure 27 (Supplementary Note 10). For PEABr films with and without crown, the decays up to moderately high excitation density are characterized by a single lifetime, consistent with a recombination process that is independent of excitation density. Only at very high excitation density above approximately 10$^{18}$ cm$^{-3}$ the lifetime decreases, consistent with the onset of additional, higher-order recombination. Luminescence decays are not pure mono-

exponential, indicating a range of lifetimes are present in the sample, consistent with a range of crystal/domain sizes.

TA measurements (Fig. 4d–f) confirm the shift to mono-molecular recombination for OHP samples upon PEABr addition. TA directly measures the change in excited state population ($dn/dt$) as a function of excited state density $n$ by resolving population kinetics. TA characterizes all of the processes which leads to a change of carrier population, not just radiative recombination. Initial carrier density is calculated from the number of photons absorbed per excitation pulse, and tracked in time using the integral of the associated ground-state bleach[11]. As shown in Fig. 4d–f, all plots exhibit a transition from first-order recombination (slope 1) at low carrier density to third-order recombination (slope 3) at high carrier density, which occurs at lower carrier densities in films with (Fig. 4e) 40% PEABr and (Fig. 4f) 40% PEABr-crown. We note that the $PL_0$ (Fig. 4b, c) shows an increase in this regime, which is most pronounced for 40% PEABr-crown. This highlights the reduced impact of Auger recombination on radiative recombination in the treated films, which allows for the presented high PL and EL yields. In addition, $CsPbBr_3$ (0% PEABr) exhibits clear second-order recombination (slope 2) between $10^{16}$ and $10^{18}$ cm$^{-3}$ which we assign to a regime in which radiative bimolecular recombination is dominant. This regime is largely absent in the films containing PEABr. $CsPbBr_3$ (0% PEABr) therefore exhibits recombination kinetics typical of bulk perovskite, for which radiative recombination is a non-geminate process, whereas PEABr films (with or without crown) exhibit recombination consistent with nano-crystalline perovskite, for which radiative recombination is a geminate-pair process.

We extract the recombination rate constants by fitting the data in Fig. 4d–f using Eq. (1).

$$-\frac{dn}{dt} = k_1 \cdot n + k_2 \cdot n^2 + k_3 \cdot n^3. \quad (1)$$

The fitted rate constants are presented in Table 1, while the fitted functions are presented in Supplementary Figure 28. The fitted second-order recombination coefficient ($k_2$) decreases substantially from $1 \times 10^{-9}$ s$^{-1}$ (0% PEABr) to $5 \times 10^{-11}$ s$^{-1}$ (40% PEABr) and $2 \times 10^{-1}$ s$^{-1}$ (40% PEABr-crown) at room temperature. For films containing PEABr, reasonable fits to experimental data may be achieved by setting $k_2$ to zero (Supplementary Figure 28). These results support the above discussion of the microstructure and device physics: The addition of crown leads to enhanced (geminate) radiative recombination and reduced non-radiative recombination losses due to a smaller and more uniform size distribution of nano-crystalline OHP domains with spatial confinement of charge carriers. We have obtained direct evidence from transient absorption spectroscopy (Supplementary Figures 29 and 30) that in the presence of 40% PEABr, both with and without crown, efficient energy funneling occurs on a ps time scale within the distribution of nanocrystal in the films[4,5].

## Discussion

In conclusion, we report an OHP LED with 15.5% EQE, which is achieved by additive-based tuning of film structure and recombination pathways. Our strategy can also benefit other perovskite systems, such as the recent mixed-dimensionality OHP solar cells, and allow better control of phase separation and crystallization through the incorporation of judiciously selected organic molecules.

## Methods

**Materials.** $PbBr_2$ (99.999%, metals basis), CsBr (99.999%, metals basis), and HBr (99.9999% metals basis, 48 wt% in water) were purchased from Alfa Aesar. PEA (99%), 1,4,7,10,13,16-hexaoxacyclooctadecane (18-crown-6, crown) (99%) and chlorobenzene (99.8% extra dry) were purchased from Acros. Dimethyl sulfoxide (DMSO, 99.9%) was purchased from Innochem. Poly [bis (4-phenyl) (4-butyl-phenyl) amine] (poly-TPD) was purchased from American Dye Source. 2′,2′-(1,3,5-benzinetriyl)-tris(1-phenyl-1-H-benzimidazole)) (TPBi) was purchased from Han Feng Chemical Co. Poly[(9,9-bis(3′-(N,N-dimethylamino)propyl)-2,7-fluor-ene)-alt-2,7-(9,9-dioctylfluorene)] (PFN) was purchased from 1-Materials. All the chemical materials were directly used without any further purifications.

**Synthesis of PEABr.** PEABr was synthesized by adding 6.71 g hydrobromic acid to a solution of phenethylamine in ethanol (anhydrous, 25 ml) with vigorously stir-ring at 0 °C for 2 h. The PEABr precipitate was obtained by evaporating the solution at 50 °C, which was washed by ethanol for three times and then dried under vacuum at 40 °C for 24 h.

**Preparation of perovskite films.** Perovskite films were obtained by spin-coating precursor solutions onto substrates. The molar ratio of $x$% PEABr meant $M_{PEABr}/M_{CsPbBr_3} = x\%$. The precursor solutions were obtained by mixing 0.2 mmol CsBr and 0.2 mmol $PbBr_2$ in DMSO with different amounts of PEABr at 80 °C for 2 h with constant stirring. Unless specially mentioned, the crown concentration in perovskite precursor is 3.5 mg ml$^{-1}$ with a mole ratio of $CsBr:PbBr_2$:crown is 1:1:0.07. Two-step process was used during spin-coating perovskite pre-cursor onto the substrates (1000 r.p.m. for 5 s and 3000 r.p.m. for 55 s, respec-tively). Finally, the resulting film was annealed at 100 °C for 1 min to accelerate crystallization.

**Device fabrication.** All the precursor solution was filtered by 0.45 μm hydrophobic poly(tetrafluoroethylene) syringe filters before using. And all the processes were carried out in a nitrogen-filled glove box. Poly-TPD (10 mg ml$^{-1}$ in chlor-obenzene) was spin-coated on a cleaned ITO substrate at 1000 r.p.m. for 60 s, followed by annealing at 150 °C for 20 min. In order to improving the wettability, a very thin PFN film (<5 nm) was spin-coated on poly-TPD layer. After that, the perovskite film was deposited on PFN layer. TPBi (20 nm), LiF (1 nm), and Al (100 nm) were deposited by thermal evaporation with vacuum pressure below $2 \times 10^{-6}$ mbar, respectively. The active device area is 0.09 cm$^2$ except notified. Finally, the devices were sealed by ultraviolet-curable resin in a nitrogen-filled glove box before testing. The device electrical output characteristics were measured in ambient air condition.

**Film characterization and device measurements.** UV–Vis absorption spectra of perovskite film was obtained by a UV-vis spectrometer (SPECORD S 600). PL spectra and PLQY were acquired by IHR 320 (Horiba Instruments Inc.). PL decay lifetimes were collected by a fluorescence spectrophotometer (HORIB-FM-2015). The excitation intensity is measured by ThorLabs PM100D. DLS measurement is carried by Nano–ZS90 (Malvern). SEM images of perovskite film and cross-section were measured with a Carl Zeiss Supra 55. SEM-EDX was measured by Gemini 500 and Oxford Xmax 20. AFM images were measured in tapping mode with an Asylum Research Cypher S AFM microscope. XRD measurements were performed with a Bruker D8 Advance X-ray diffractometer. The GIXRD measurements were performed at the BL14B1 beamline of the Shanghai Synchrotron Radiation Facility using X-ray with a wavelength of 1.24 Å. 2D GIXRD patterns were acquired by a MarCCD mounted vertically at a distance of approximately 326 mm from the sample with a grazing incidence angle of 0.2° and an exposure time of 50 s. The 2D GIXRD patterns were analyzed using the FIT2D software and displayed in scat-tering vector $q$ coordinates. The angle-dependent emission profile of perovskite LED device was measured by Hamahatsua C9920-11. Keithley 2400 sourcemeter and a PhotoResearch spectrometer PR670 were used for $J$–$V$–$L$ characteristics and EL spectra of perovskite-based LEDs, respectively. A scan rate used for $J$–$V$–$L$ characteristics was approximately 5000 ms for each data point. $^1$H NMR spectra were recorded in dimethyl sulfoxide-d6 (DMSO-d6) on a Bruker 400 MHz NMR spectrometer at room temperature.

**EQE calculation method.** EQE is a number of photons generated by the LED device per second ($N_{photon}(V)$) divided by a number of charges injected to the

**Table 1 Fitted recombination rate constants from TA measurement for perovskite films of different composition**

| Films | $k_1$ (s$^{-1}$) | $k_2$ (cm$^3$ s$^{-1}$) | $k_3$ (cm$^6$ s$^{-1}$) |
|---|---|---|---|
| 0% PEABr | $4 \times 10^{-6}$ | $1 \times 10^{-9}$ | $5 \times 10^{-28}$ |
| 40% PEABr | $6 \times 10^{-6}$ | $5 \times 10^{-11}$ | $4 \times 10^{-28}$ |
| 40% PEABr-Crown | $9 \times 10^{-6}$ | $2 \times 10^{-10}$ | $2 \times 10^{-27}$ |

device per second ($I(V)/e$)[4]:

$$\mathrm{EQE}(V) = \frac{N_{\mathrm{photon}(V)}}{I(V)/e} \times 100\%, \qquad (2)$$

where $I(V)$ is a current (unit: A) passing through the perovskite LED device at an applied bias ($V$), $N_{\mathrm{photon}(V)}$ represents the number of emitted photons per second gathered. $e$ is elementary charge of $1.6 \times 10^{-19}$ C.[24]

The $N_{\mathrm{photon}(V)}$ is calculated by

$$N_{\mathrm{photon}(V)} = \frac{\Phi_e}{E_{\mathrm{average}} \times 1.6 \times 10^{-19}}, \qquad (3)$$

where $\Phi_e$ is radiant flux (W). $E_{\mathrm{average}}$ (eV) means the average photon energy among the whole EL spectrum at a bias. The units between them are different. So, the conversion factor ($1.6 \times 10^{-19}$ J eV$^{-1}$) is needed for calculation.

There are two different concepts in photometry, radiant flux ($\Phi_e$, W), and luminous flux ($\Phi_v$, lm). The radiant flux contains the total power of electromagnetic radiation, while the luminous flux is that adjusted to reflect the varying sensitivity of the human eye to different wavelengths of light. The relationship between $\Phi_e$ and $\Phi_v$ is given by[25]

$$\Phi_v = K_m \int \Phi_{e,\lambda} V(\lambda) \mathrm{d}\lambda, \qquad (4)$$

where $V(\lambda)$ is the luminosity function, representing the average spectral sensitivity of human visual perception of brightness. $\lambda$ is the wavelength (nm). $K_m$ is a constant, 683 lm W$^{-1}$ at 555 nm.

So, $\Phi_e$ is given by

$$\Phi_e = \int \frac{\Phi_{v,\lambda}}{K_m V(\lambda) \mathrm{d}\lambda}. \qquad (5)$$

Here, the perovskite LED can be assumed a Lambertian radiator according to its angular intensity light distribution profile, and the device shows uniform emission in over 1 cm², as shown in Fig. 3d and Supplementary Figure 20b. So $\Phi_v = \pi AL$, where $\pi$ is the solid angle; $A$ is the active area (m²) of a working LED device; $L$ is the luminance (cd m$^{-2}$) measured by PR670, as shown in Supplementary Figure 21. Then $\Phi_e$ is given by

$$\Phi_e = \frac{\pi AL}{K_m \int V(\lambda) \mathrm{d}\lambda}. \qquad (6)$$

The relationship between the photon energy ($E_{\mathrm{average}}$, eV) and the photon wavelength ($\lambda$, nm) is photon wavelength (nm) = 1240/photon energy (eV). Then, $E_{\mathrm{average}}$ is calculated by the following equation:

$$E_{\mathrm{average}} = \frac{\int F(\lambda) \frac{\lambda}{1240} \mathrm{d}\lambda}{\int F(\lambda) \mathrm{d}\lambda} \qquad (7)$$

$F(\lambda)$ is the photon radiometric value (W sr$^{-1}$ m$^{-2}$) collected by PR670, as shown in Supplementary Figure 21.

**Transient characterization**. Time-resolved PL measurements were conducted with a gated-intensified CCD camera system (Andor iStar DH740 CCI-010) integrated with a grating spectrometer (Andor SR303i). Femtosecond excitation laser pulses were generated in a home-built setup. The second harmonic of the output from a Ti:Sapphire laser system (Spectra Physics Solstice) was generated in a beta barium borate (BBO) crystal (pulse energy 1.55 eV, pulse length 80 fs) and used as excitation pulses. Pump scatter from the excitation laser was removed by an absorptive long-pass filter with a 425 nm edge (Thorlabs).

Transient absorption was carried out with the output of a Ti: Sapphire amplifier system (Spectra Physics Solstice) operating at 1 kHz and generating 90 fs pulses, split into pump and probe beam paths. Visible broadband probe beams were generated in home-built noncollinear optical parametric amplifiers. The ps measurements were carried out with a 400 nm narrowband (10 nm full-width at half-maximum) pump beam provided by a TOPAS optical parametric amplifier (Light Conversion). The ns measurements were carried out with the frequency-tripled (355 nm) output of an electrically triggered q-switched pulsed laser (Advanced Optical Technologies AOT-YVO-25QSPX) pump. The transmitted pulses were collected with an InGaAs dual-line array detector (Hamamatsu G11608-512) driven and read out by a custom-built board from Stresing Entwicklungsbüro.

## Data availability

The experimental data that support the findings of this study are available from the corresponding author on request.

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

## Acknowledgements

This work was supported by the National Key Research and Development Program of China (2016YFA0202402), the National Natural Science Foundation of China (91123005, 61674108, 11550110176, 11675252, 11605278), the Priority Academic Program Development of Jiangsu Higher Education Institutions (PAPD), the 111 Project and Collaborative Innovation Center of Suzhou Nano Science and Technology (NANO-CIC).

## Author contributions

M.B. and Y.Z. designed the materials and device structure, and fabricated the LEDs. M.B. performed the AFM, XRD, DLS, PL, and absorption measurements. Y.Y and X.G.

assisted with synchrotron GIXRD characterization. J.P.H.R. conducted transient PL and TA characterization, and T.T. assisted absorption characterization. Y.T. synthesized QDs and fabricated the QD-based LED. T.S. assisted LED characterization. F.D., H.S., D.C. and B.S. supervised the work. M.B., H.S. and B.S. wrote the manuscript. All authors read and commented on the manuscript.

## Additional information

**Competing interests:** The authors declare no competing interests.

