## [Peer Review File · Nature Communications]

Editorial Note: This manuscript has been previously reviewed at another journal that is not operating a transparent peer review scheme. This document only contains reviewer comments and rebuttal letters for versions considered at Nature Communications. With redaction in the interest of confidentiality.

Reviewers' comments:

Reviewer #1 (Remarks to the Author):

The authors report high efficient perovskite LEDs with max. EQE 15.5 % and improved photoluminescence quantum yield of 70 % by precisely controlling organometal halide perovskite crystallite distribution and phase segregation. The authors can control the crystallite by adding PEABr to CsPbBr₃. Especially, the authors discovered that using a crown as an additive prevents aggregation between PEABr and improves device performance. Electrical and optical characterization are well organized and are in good agreement with experimental results.

1. It would be better to show the transient absorption spectra at different timescales showing that the relative intensity of the each bleaching peaks to confirm that the energy funnel effect occurs.
2. Also, when the crown was added, the optical absorption data showed that the perovskite peak below 500 nm decrease. It is better to show the transient absorption spectra to confirm that energy funneling still occurs.
3. Your devices have the similar structure with the previously-reported devices (H. Cho, T.-W. Lee et al., Science 2015 350, 1222. / L. Zhao, B. P. Rand et al., ACS Nano 2017 11, 3957.). However, the emission profile was quiet deviated from the Lambertian and the reported emission profile. Can you include the reason for the such deviation?
4. The pixel size was larger than the previously reported pixel size. If the pixel is large, it may not be uniform. So, it is necessary to confirm that even small pixel still shows the same level of good performance (0.2cm x 0.2cm).
5. It would be better to include the hysteresis results(forward, backward) to check for ion migration.
6. It is necessary to explain how the figure S7 was derived from the TEM image of 50 nm scale. To illustrate Figure S7, you need the data observed with a smaller scale TEM image.

Reviewer #2 (Remarks to the Author):

The authors have made large efforts to take in account the criticisms of the reviewers, while the manuscript became definitely more precise in the description of the experiments I have still the feeling that has a rather phenomenological touch and that the role of crown is not really explained - or at least it is not explain while crown it is so special. However, I agree that the results, in terms of device performances are really excellent and therefore the manuscript has in this respect a value and interest for the community. I therefore suggest its acceptance in the current form.

Reviewer #3 (Remarks to the Author):

I have read the manuscript and the responses made by the authors in response to the earlier review. Unfortunately, I do not think the work merits publication in Nature Communication. There are many reports of high efficiency LED now out, and a minor improvement in efficiency with a "quirky" recipe is of less utility to the community without a clear physical model being established.

In terms of novelty, while the addition of PEABr into CsPbBr₃ to create insitu nanoparticle film is not novel, I agree that the addition of crown ether to the system to minimize the PEABr segregation into the system might be something that has not been reported before. However, the mechanism as to how crown ether help to minimize PEABr segregation is not fully explored and it is based on hypothesis only.

Moreover, the evidence which indicate minimum PEABr segregation with crown ether addition is not solid. For example : Figure S7 (TEM of drop casted film) was used as an evidence however, the author does not show EDX to prove where is the PEABr or Cs rich region. It is based on visual observation in which one is smoother than the other. The absorption curve (Figure S9B) also indicates reduction in the overall absorption curve with crown ether addition despite similar background absorbance, indicating thinner film which might affect the XRD and GIXRD analysis as well since the provided evidence for less PEABr segregation is obtained from smaller XRD peak of PEABr. The differentiation between crown ether reduced the PEABr + CsPbBr₃ formation in general as compared to less PEABr segregation in the film is not clear here.

Also, the choice of crown ether should be explained, as the author's mention that the cavity size of the 18-Crown-[6] is 130-160 pm, while Cs⁺ is 169 pm (it does not fit, or at least lower affinity). Moreover, a bivalent cation (Pb²⁺) is more likely to form a complex with the crown ether than a monovalent cation (Cs⁺). Figure S9a, for example, shows that the C.E. interacts with the PbBr₂, but the effect of CsBr cannot be explained when the mixture of CsBr+PbBr₂ is measured (inconclusive experiment) as CsBr displays no signal in the absorption spectra). More detailed analysis is therefore required (e.g. NMR) to prove which component forms a complex: PbBr₂, CsBr, or both. The discussion on the role of the crown ether, and its effect on e.g. growth suppression, should therefore be adjusted/extended. In addition, the author argued that additional crown ether on 40% and 60% PEABr samples increase the binding energy which therefore redshifted the PL. However, this argument is in conflict with the fact that blue shifted PL was noticed when dielectric confinement was introduced with increasing PEABr in the sample as well. Moreover, blue shifted PL was observed with crown ether addition on 0% PEABr. How could the author explain this ? Also, no EL shift (Figure 18c) was observed on devices with and without crown ether as contrast to the PL measurement. Could the author explain this ?

As addition of crown into 0% PEABr also reduces the crystallite size, it is clear that there are two competing mechanisms that led to smaller crystallite. As crown is also organic molecule(with 7% addition), it is fair to assume crown might also contribute to the dielectric confinement. The author ought to verify the optical properties (PLQY, TRPL, and TA, binding energy etc) and LED performance with 0% PEABr + crown.

The author cites "technical" problem with 0.5 V measurement step in the rebuttal, it still does not answer the various scan rate measurement as requested by previous referee. Nevertheless, authors should still provide a better quality data for publication since the title clearly highlights the high efficiency. The question regarding the reproducibility was also ignored. Please see data presented in Nature Communications volume 9, Article number: 570 (2018) doi:10.1038/s41467-018-02978-7

Probably typo: Fig S12, while caption indicate 3D CsPbBr₃, the legend mentions PEABr.

Reviewer #1 (Remarks to the Author):

The authors report high efficient perovskite LEDs with max. EQE 15.5 % and improved photoluminescence quantum yield of 70 % by precisely controlling organometal halide perovskite crystallite distribution and phase segregation. The authors can control the crystallite by adding PEABr to CsPbBr₃. Especially, the authors discovered that using a crown as an additive prevents aggregation between PEABr and improves device performance. Electrical and optical characterization are well organized and are in good agreement with experimental results.

Comment 1#. It would be better to show the transient absorption spectra at different timescales showing that the relative intensity of the each bleaching peaks to confirm that the energy funnel effect occurs.

Reply: We thank the referee for this suggestion, and we present a detailed analysis of transient absorption experiments in Supplementary Fig. S29 and S30 to investigate energy funneling. As requested, we show transient absorption spectra at different time delays for CsPbBr₃ films with and without PEABr/PEABr-Crown. As can be seen from the spectral shifts and kinetics in Supplementary Fig. 30 and Fig. 31, there is clear energy relaxation and evidence for energy funneling in films with PEABr/PEABr-Crown, while only carrier cooling is observed for pristine CsPbBr₃ films.

Revision: In the revised manuscript, a sentence of “We have obtained direct evidence from transient absorption spectroscopy (Supplementary Fig. 29 and 30) that in the presence of 40% PEABr, both with and without crown, efficient energy funneling occurs on a ps time scale within the distribution of nanocrystallites in the film^{4,5}.” is added.

Comment 2#. Also, when the crown was added, the optical absorption data showed that the perovskite peak below 500 nm decrease. It is better to show the transient absorption spectra to confirm that energy funneling still occurs.

Reply: We show the transient absorption spectra for these films in Supplementary Fig. 29 and 30 – please see our reply to Comment 1#. Energy funneling occurs on ultrafast timescales within the first 2ps after excitation in films with and without crown additive, which shows that excitations transfer efficiently to the nanocrystallites with low bandgaps.

Revision: Please see our revision to comment 1#.

Comment 3#. Your devices have the similar structure with the previously-reported

devices (H. Cho, T.-W. Lee et al., *Science* 2015 350, 1222. / L. Zhao, B. P. Rand et al., *ACS Nano* 2017 11, 3957.). However, the emission profile was quite deviated from the Lambertian and the reported emission profile. Can you include the reason for the such deviation?

Reply: It is well known that the refractive index of each layer in a light emitting diode affects emission profile. In previous reported perovskite LED devices, a structure of glass/ITO/PEDOT:PSS/PFI/MAPbBr₃/TPBi/LiF/Al (*Science*, 2015, 350, 1222) or glass/ITO/Poly-TPD/MAPbBr₃ (MAPbI₃)/TPBi/LiF/Al (*ACS Nano*, 2017, 11, 3957) is used. If the active layer is different, the emission profile is quite different, **REDACTED** (*ACS Nano*, 2017, 11, 3957). With incorporating organic ligand in perovskite, the refractive index is changed, which should result in a variation in the emission profile **REDACTED** (glass/ITO/Poly-TPD/MAPbI₃:organic ligand/TPBi/LiF/Al, *Nat. Photon.* **2017** 11, 108). In previous reported work, the *Rand* group has pointed out that the emission profile is significantly deviates from the Lambertian one for LEDs based on I-perovskite LEDs with different BA1:MAPbI₃ molar ratio, **REDACTED**. Thus, we attribute these emission profile deviations to differences in the refractive index of the perovskite layers as well as slight differences of device structure.

REDACTED

REDACTED

Comment 4#. The pixel size was larger than the previously reported pixel size. If the pixel is large, it may not be uniform. So, it is necessary to confirm that even small pixel still shows the same level of good performance (0.2cm x 0.2cm).

Reply: In the previous report, the active device area was generally in small size (0.1 cm², *Nat. Photon.* 2017 11, 108; 0.03 cm², *Nat. Photon.* 2016, 10, 699; 0.0614 cm² *Nat. Nanotechnol.*, 2016, 11, 872). Here, in our work, the active device area was 0.09

cm², which was comparable with previous ones. To test the uniformity, a LED device with an emitting size of 1.5×1 cm² was fabricated. The device photograph operated at a bias of 3V and 4 V was shown in Fig. 3d and Supplementary Fig. 20b, respectively. Both of them showed relatively uniform large-area emission. From these measurements there is no evidence for non-uniform emission.

Revision: This has been clarified on page 13 of the manuscript.

5. It would be better to include the hysteresis results (forward, backward) to check for ion migration.

Reply: The forward and backward J-V-L measurements were conducted according to the referee's suggestion. As shown in Supplementary Fig. 23, it can be observed that some level of hysteresis, i.e. a difference in response for forward and backward scans, occurs for both J and L. However, the level of hysteresis is relatively minor and comparable with similar device in the literature (*Nat. Photon.*, **2017**, 11, 108). A tentative assignment for this hysteresis should be ion migration and associated traps in the device. With incorporation of crown, the 40% PEABr-crown device exhibits further reduced hysteresis. We believe that this is consistent with the more efficient carrier confinement with first-order recombination characteristics and suppression of non-radiative recombination which may be manifestations of a reduced trap density in the perovskite film.

Revision: In the revised manuscript a discussion is added on page 13 of the main text and Supplementary Fig. 23 is added.

6. It is necessary to explain how the figure S7 was derived from the TEM image of 50 nm scale. To illustrate Figure S7, you need the data observed with a smaller scale TEM image.

Reply: TEM images with smaller scales are shown in inset figures, as shown in Supplementary Fig. 7. With incorporation of crown, the organic shell thickness surrounding perovskite nanocrystal becomes much thinner. PEABr aggregation and phase separation can be easily observed.

Revision: In the revised manuscript, the inset figures are added in Supplement Fig. 7.

Reviewer #2 (Remarks to the Author):

The authors have made large efforts to take in account the criticisms of the reviewers, while the manuscript became definitely more precise in the description of the experiments I have still the feeling that has a rather phenomenological touch and that the role of crown is not really explained - or at least it is not explain while crown it is so special. However, I agree that the results, in terms of device performances are really excellent and therefore the manuscript has in this respect a value and interest for the community. I therefore suggest its acceptance in the current form.

Reply: Many thanks for the reviewer's positive comments.

Regarding the crown function, further experiment such as ^1H NMR, SEM-EDX, TA, TEM, EL and TSPC, all the data strongly support that the presence of crown dramatically suppresses the phase segregation between PEABr and inorganic perovskite.

Reviewer #3 (Remarks to the Author):

Comment 1#: I have read the manuscript and the responses made by the authors in response to the earlier review. Unfortunately, I do not think the work merits publication in Nature Communication. There are many reports of high efficiency LED now out, and a minor improvement in efficiency with a “quirky” recipe is of less utility to the community without a clear physical model being established.

In terms of novelty, while the addition of PEABr into CsPbBr₃ to create insitu nanoparticle film is not novel, I agree that the addition of crown ether to the system to minimize the PEABr segregation into the system might be something that has not been reported before. However, the mechanism as to how crown ether help to minimize PEABr segregation is not fully explored and it is based on hypothesis only.

Moreover, the evidence which indicate minimum PEABr segregation with crown ether addition is not solid. For example: Figure S7 (TEM of drop casted film) was used as an evidence however, the author does not show EDX to prove where is the PEABr or Cs rich region. It is based on visual observation in which one is smoother than the other. The absorption curve (Figure S9B) also indicates reduction in the overall absorption curve with crown ether addition despite similar background absorbance, indicating thinner film which might affect the XRD and GIXRD analysis as well since the provided evidence for less PEABr segregation is obtained from smaller XRD peak of PEABr. The differentiation between crown ether reduced the PEABr + CsPbBr₃ formation in general as compared to less PEABr segregation in the film is not clear here.

Reply: In order to provide more direct evidence of phase separation, we have performed scanning electron microcopy-energy-dispersive X-ray spectroscopy (SEM-EDX) mapping, as shown in Supplementary Fig. 8. After incorporating crown in 40% PEABr perovskite film, the distributions of Pb, Cs and N become more uniform than 40% PEABr perovskite film, providing evidence for less pronounced phase separation.

In Supplementary Fig. 9b, the overall absorption is reduced with the addition of crown. However, the UV-vis absorption spectrum changed sharply only at high concentration of crown (>30%). In our optimized LED device, the concentration of crown is only 7% which is much lower than 30%. The addition of crown only slightly affects the thickness of perovskite films (0% PEABr: 34.56±0.58 nm, 0% PEABr-crown: 37.20±0.32 nm; 40% PEABr: 39.46±0.33 nm, 40% PEABr-crown: 42.65±0.15 nm; 60% PEABr: 45.13±0.46 nm; 60% PEABr-crown: 50.01±0.43 nm). It shows ~10% difference before and after crown addition. These relatively minor differences in thickness cannot explain the large differences of then PEABr diffraction signals in XRD and GIXRD.

Revision: In the revised manuscript, the sentences “In order to provide the further evidence...” are added in Supplementary information (ST2). Supplementary Fig. 8 is added.

Comment 2#: Also, the choice of crown ether should be explained, as the author’s mention that the cavity size of the 18-Crown-[6] is 130-160 pm, while Cs⁺ is 169 pm (it does not fit, or at least lower affinity). Moreover, a bivalent cation (Pb²⁺) is more likely to form a complex with the crown ether than a monovalent cation (Cs⁺). Figure S9a, for example, shows that the C.E. interacts with the PbBr₂, but the effect of CsBr cannot be explained when the mixture of CsBr+PbBr₂ is measured (inconclusive experiment) as CsBr displays no signal in the absorption spectra). More detailed analysis is therefore required (e.g. NMR) to prove which component forms a complex: PbBr₂, CsBr, or both. The discussion on the role of the crown ether, and its effect on e.g. growth suppression, should therefore be adjusted/extended.

Reply: To further confirm the interaction between crown and PEABr, CsBr, PbBr₂, we use ¹H NMR spectra to characterize the interaction between crown and Pb²⁺, Cs⁺, PEA⁺. ¹H NMR spectra of crown with Pb²⁺, Cs⁺, PEA⁺ individually are measured, respectively. As shown in Supplementary Fig. 14, the proton resonance signals of crown ether (peak at $\delta=3.506$ p.p.m.) downfield chemical shift towards downfield after incorporating CsBr (3.529 p.p.m.), PbBr₂ (3.531 p.p.m.) and PEABr (3.547 p.p.m.). This observation is consistent with previous reports on downfield chemical shift of proton in crown ether for ¹H NMR spectra because of the interaction between cation and crown ether (*J. Am. Chem. Soc.*, 1976, 98, 3769). Here, in our work, such chemical shift can be assigned to a hydrogen bond between PEA⁺ and the oxygen atom in crown, or a coordinating bond between Pb²⁺, Cs⁺ and crown. These bond interactions result in proton chemical environmental change. This downfield chemical shift for proton in crown increases from Cs⁺, to Pb²⁺ and PEA⁺, which reveals that the interaction between crown and PEA⁺ is the strongest.

This observed chemical shifts are consistent with the radius of Pb²⁺ (120 pm) being more suitable for crown (18-crown-6, hole radius is 130-160 pm) than Cs⁺ (169 pm), from which one would expect a stronger interaction between Pb²⁺ and 18-crown-6 than that between Cs⁺ and 18-crown-6. There are three hydrogen bonds between PEA⁺ and crown, so the interaction is expected to be the strongest among them. Besides that, the proton resonance signals of PEA⁺ (peak at $\delta=7.893$ p.p.m.) shift towards upfield after incorporating with crown as shown in Supplementary Fig. 14, which can also be attributed to the hydrogen bonds discussed above.

In order to investigate the interaction in mixed systems, we prepare two samples, PEABr/crown/PbBr₂(molar ratio is 1:1:1) and PEABr/crown/PbBr₂/CsBr (molar ratio is 1:1:1:1). The ¹H NMR spectra for these two samples are also shown in Supplementary Fig. 14. When there are three components (PEABr/crown/PbBr₂) in

solution, the shifts of proton resonance signals of PEABr and crown is similar to that of PEABr/crown (without PbBr_2). This provides further evidence that PEABr and crown exhibit stronger interaction than PbBr_2 and crown. However, when crown, Pb^{2+} , Cs^+ , and PEA^+ co-exist in a solution, the behavior is more complex and the chemical shifts become smaller. We speculate that this is because perovskite nanocrystals exist even in the solution, which would reduce the interaction between crown and these three ions.

In summary, the interactions between crown and Pb^{2+} , Cs^+ , PEA^+ are proven by ^1H NMR. And the comparative strength of the interactions of crown is in an order of $\text{PEA}^+ > \text{Pb}^{2+} > \text{Cs}^+$.

Revision: In the revised manuscript, three paragraphs are added in ST4 of Supplementary Information. Supplementary Fig. 14 is added.

Comment 3#: In addition, the author argued that additional crown ether on 40% and 60% PEABr samples increase the binding energy which therefore redshifted the PL. However, this argument is in conflict with the fact that blue shifted PL was noticed when dielectric confinement was introduced with increasing PEABr in the sample as well. Moreover, blue shifted PL was observed with crown ether addition on 0% PEABr. How could the author explain this? Also, no EL shift (Figure 18c) was observed on devices with and without crown ether as contrast to the PL measurement. Could the author explain this?

Reply: To explain this we need to consider that both quantum confinement (blue shift) and dielectric confinement (red shift) can affect the PL spectrum, and they are competitive mechanisms affecting PL peak position. If quantum confinement (crystal size effect) is dominated, the PL spectra would show blue shift. With the addition of PEABr, the crystal size dramatically decreases. As shown in Supplementary Tab. 1, the average crystallite sizes decrease sharply from 42.8 nm (0% PEABr) to 18.5 nm (40% PEABr). If the dielectric confinement is dominant, the red shift in PL spectra would occur. Here, in 40% PEABr perovskite without and with crown, the average crystal size changes from 18.5 nm (without) to 13.3 nm (with). The average crystal size is only slightly reduced, and this size is much larger than its Bohr diameter. So we speculate that here the dielectric confinement is the dominant factor, which results in weak PL redshift, as shown in Fig. 2b for 40% PEABr-crown perovskite films. This PL redshift is even more obvious for 60% PEABr-crown perovskite film, as shown in Fig. 2b, where PEABr is uniformly distributed around perovskite and dielectric confinement is enhanced.

In the 40% PEABr LED (Supplementary Fig. 19c) an EL shift upon addition of crown was indeed too small. However, in 60% PEABr samples an EL redshift upon addition of crown was clearly detected. As shown in Supplementary Fig. 19d, the EL spectrum red shifts from 508 nm to 510 nm in the presence of crown. This is consistent with

the PL measurements, although one has to consider of course that one cannot expect a 1:1 correspondence as EL and PL sample different crystallite distributions.

Regarding the 0% PEABr, as shown in Supplementary Fig. 11 and Fig. 12, the average crystallite sizes decrease from 45.8 nm (0% PEABr) to 20.3 nm (0% PEABr+ 7% crown). With such a dramatic change in crystallite size, we assume that the observed blue-shift in the crown samples reflects primarily the smaller size of the lowest bandgap nanocrystallites in the distribution.

Revision: In the revised manuscript, Supplementary Fig. S19d is added.

Comment 4#: As addition of crown into 0% PEABr also reduces the crystallite size, it is clear that there are two competing mechanisms that led to smaller crystallite. As crown is also organic molecule (with 7% addition), it is fair to assume crown might also contribute to the dielectric confinement. The author ought to verify the optical properties (PLQY, TRPL, and TA, binding energy etc.) and LED performance with 0% PEABr + crown.

Reply: We totally agree that there two competing mechanisms where both PEABr and crown can suppress the crystal growth. In the sample for 40% PEABr-crown system, only 7% crown (mole ratio to PbBr_2) is added, and this limited amount of crown can only play a weak effect on suppressing perovskite growth. In 40% PEABr perovskite without and with crown, the average crystal size changed from 18.5 nm (without) to 13.3 nm (with) and the PL intensity dramatically increased (Fig. 2c). With crown addition in 0% PEABr sample, the average crystal sizes decrease from 45.8 nm (without) to 20.3 nm (with), while the PL intensity is only slightly enhanced (Supplementary Fig. 9c, d). The PL intensity for 0% PEABr sample (with or without crown) is too weak to allow integrating sphere measurements to probe an accurate PLQY value, and the PL lifetime slightly increased with incorporation with crown (Supplementary Fig. 9e). The enhanced PL intensity for 0% PEABr-crown perovskite should be ascribed to size-effect confinement. This suggests that dielectric confinement contributions for crown would be comparatively weak in comparison with PEABr. Otherwise, much enhanced PL intensity should be observed in 0% PEABr-crown perovskite. In addition, the morphology of 0% PEABr-crown film is still very poor (Supplementary Fig. 11b), all the LED devices are shorted and we unable to report LED performance data for this composition.

Revision: In the revised manuscript, a paragraph is added in ST3 of Supplementary Information. Supplementary Fig. 9c, d, e are added.

Comment 5#: The author cites “technical” problem with 0.5 V measurement step in the rebuttal, it still does not answer the various scan rate measurement as requested by previous referee. Nevertheless, authors should still provide a better quality data for publication since the title clearly highlights the high efficiency. The question

regarding the reproducibility was also ignored. Please see data presented in Nature Communications volume 9, Article number: 570 (2018) doi:10.1038/s41467-018-02978-7

Reply: We measure LED devices with different measurement bias step (0.75, 0.5, 0.25 and 0.1 V), and a constant time at each step is 5 s. This is limited by a minimum exposure time of PhotoResearch spectrometer PR670. Therefore, the scan rate of 0.75, 0.5, 0.25, 0.1 V step is 0.15, 0.1, 0.05 and 0.02 V/s, respectively. It takes 40, 60, 120 and 300 s to measure a device from 0 V to 6V when bias step is 0.75, 0.5, 0.25 and 0.1 V, respectively. The results are shown in Supplementary Fig. 24. For the 0.75 V measurement step, the current efficiency and luminance are the highest among all of measurement step because of taking the shortest measuring time (40 s). And with the decreased measurement step, the LED device performance becomes less stable. Especially, the current efficiency and luminance decays with 0.1 V measurement step (0.02 V/s) because of taking the longest measurement time (300 s). A tentative assignment for the device quick degradation is joule heat when the step size of the measurement is too small. We do find that PL intensity is significantly degraded during long time thermal annealing even at 100 °C. Here, in order to balance device efficiency and reliability, 0.5 V bias step (0.1V/s) is used.

Revision: Three sentences of “Supplementary Fig. 24 shows the scanning-rate-dependent current efficiency for 40% PEABr-crown device...” are added in the manuscript. In the revised manuscript, Supplementary Fig. 24 is added.

Comment 6#: Probably typo: Fig S12, while caption indicate 3D CsPbBr₃, the legend mentions PEABr.

Reply: It's a mistake and has been corrected in the revised version.

REVIEWERS' COMMENTS:

Reviewer #1 (Remarks to the Author):

In the manuscript titled " Solution-Processed Perovskite Light Emitting Diodes with Efficiency Exceeding 15% through Additive-Controlled Nanostructure Tailoring", the concerns of the reviewers were adequately addressed in respect of carrier dynamics on energy funneling and expected crystallite distribution. Also, supplemented experimental results about transient absorption and device hysteresis showed improved carrier dynamics and device stability with PEABr and crown ether additives, strengthening the solid insight in correlation between recombination characteristics and luminescent properties. We think the manuscript is largely strengthened after the revision, and can be accepted by Nature Communication now.

Reviewer #3 (Remarks to the Author):

The manuscript has been largely improved and the authors have responded point by point to the reviewers' questions, providing a large amount of additional data: NMR, (SEM-EDX), scanning rate dependence trends. Moreover more details on the fabrication and the samples (like the film thickness) to clarify the combined role of PEA and CE have been provided. There are still some points that are not so clear, as the surpassingly low PL signal of 0% PEA. Another weak point is the device instability (fully degrade in 300s). Taking into account that the device efficiencies are very good and the paper is vastly improved, I believe it deserves publication. The authors should comment and compare their stability performance against other perovskite LEDs before publication (in manuscript text and as a table in supplementary information).

Response to REVIEWERS' COMMENTS:

Reviewer #1 (Remarks to the Author):

In the manuscript titled " Solution-Processed Perovskite Light Emitting Diodes with Efficiency Exceeding 15% through Additive-Controlled Nanostructure Tailoring", the concerns of the reviewers were adequately addressed in respect of carrier dynamics on energy funneling and expected crystallite distribution. Also, supplemented experimental results about transient absorption and device hysteresis showed improved carrier dynamics and device stability with PEABr and crown ether additives, strengthening the solid insight in correlation between recombination characteristics and luminescent properties. We think the manuscript is largely strengthened after the revision, and can be accepted by Nature Communication now.

Reply : We thank for the referee's positive comments and for his/her previous advice on characterization funneling effect. We thank the referee for the supporting comments on the relevance and importance of our work.

Reviewer #3 (Remarks to the Author):

The manuscript has been largely improved and the authors have responded point by point to the reviewers' questions, providing a large amount of additional data: NMR, (SEM-EDX), scanning rate dependence trends. Moreover more details on the fabrication and the samples (like the film thickness) to clarify the combined role of PEA and CE have been provided. There are still some points that are not so clear, as the surpassingly low PL signal of 0% PEA.

Another weak point is the device instability (fully degrade in 300s). Taking into account that the device efficiencies are very good and the paper is vastly improved, I believe it deserves publication. The authors should comment and compare their stability performance against other perovskite LEDs before publication (in manuscript text and as a table in supplementary information).

Reply : We thank for the referee's positive comments. In addition, we really appreciate his/her advice on how to prove the complex formation of crown ether with lead bromide or cesium bromide.

Regarding to 0% PEABr based perovskite film, the low exciton binding energy as well as the poor film morphology (possible defects at the grain boundaries) is likely to dramatically reduce the photoluminescence intensity.

We measured our device stability with constant voltage and current density. Under constant voltage of 3.5 V, the champion device shows negligible efficiency off in 100 s while the device without crown decrease to its approximately 70% value (Supplementary Figure 25a), which indicates improved stability. Moreover, if the devices are measured with constant current density of 2 mA cm⁻², the operation lifetime can reach approximately 90 min (Supplementary Figure 25b), which is comparable to the others reported perovskite LEDs, as shown in Supplementary Table 4. In addition, we admit that there is still large space to improve the perovskite stability.

In the revised manuscript, we have summarized the lifetime of reported perovskite LEDs in Supplementary Table 4.